# Dominance of Influencing Factors on Cooling Effect of Urban Parks in Different Climatic Regions

**DOI:** 10.3390/ijerph192315496

**Published:** 2022-11-22

**Authors:** Songxin Zheng, Lichen Liu, Xiaofeng Dong, Yanqing Hu, Pengpeng Niu

**Affiliations:** 1College of Earth and Environmental Science, Lanzhou University, Lanzhou 730000, China; 2Key Laboratories for Environmental Pollution Prediction and Control Gansu Province, Lanzhou University, Lanzhou 730000, China; 3School of Architecture and Design, Beijing Jiaotong University, Beijing 100044, China

**Keywords:** urban park, cooling effect, influencing factors, dominance, climatic regions

## Abstract

The enhancement of the park cooling effect (PCE) is one method used to alleviate the urban heat island (UHI). The cooling effect is affected by park factors; however, the importance of these factors in the case of the PCE is still unclear. Optimizing or planning urban parks according to the importance of the influencing factors can effectively enhance the PCE. Herein, we selected 502 urban parks in 29 cities in China with three different climatic regions and quantified the PCE based on the park cooling intensity (PCI) and park cooling area (PCA). Subsequently, the relative importance of the influencing factors for the PCE was compared to identify the main factors. Consequently, certain park planning suggestions were proposed to enhance the cooling effect. The results show that: (1) the PCE increased in the order of arid/semi-arid, semi-humid, and humid regions. (2) The main factors of the PCI differed significantly in different climatic regions; however, the waterbody within a park significantly affected the PCI in all three climates. However, for the PCA, park patch characteristics were the dominant factor, contributing approximately 80% in the three climates regions. (3) In arid/semi-arid and semi-humid regions, the optimal area proportion of waterbody and vegetation within the park were approximately 1:2 and 1:1, respectively, and the threshold value of the park area was 16 ha. In contrast, in the humid region, the addition of a waterbody area within the park, to the best extent possible, enhanced the PCI, and the threshold value of the park area was 19 ha. The unique results of this study are expected to function as a guide to future urban park planning on a regional scale to maximize ecological benefits while mitigating the UHI.

## 1. Introduction

With rapid urbanization, a considerable amount of natural surface cover has been replaced by artificial surfaces, resulting in an increase in temperatures in urban areas compared to suburbs. This is known as the urban heat island (UHI) effect [1,2,3]. The UHI can cause storms and precipitation events [4]; increase energy consumption [5,6]; aggravate air pollution [7,8]; induce heat stroke, and cardiovascular-, cerebrovascular-, etc., related diseases [9,10]; and affect the livability of the urban areas [11]. However, the urban population will continue to grow in the future, and it is estimated that by 2050, the proportion of the global urban population will increase from 55% in 2018 to 68%, and from 2018 to 2050, the global urban population is expected to increase by 2.5 billion, of which China will increase 255 million [12], which will further aggravate the UHI effect. Thus, methods to reduce urban temperature and mitigate the UHI remain a crucial problem during the process of urbanization.

Urban parks, an important component of urban ecosystems, are vital to urban microclimate regulation [13]. Natural landscape elements such as vegetation, rivers, and lakes within parks can reduce the temperature of parks through shading and evapotranspiration. Subsequently, through air convection and diffusion, the cold air from parks is blown to the surrounding areas of parks to achieve regional cooling, which is considered as an important measure to alleviate the UHI [14,15]. In addition, urban parks also have many benefits, such as improving urban ventilation and reducing urban pollutant dispersion and providing leisure and entertainment places [16,17]. However, owing to the “more people and less land” nature prevalent in China, the alleviation of the UHI by increasing the number and area of urban parks is not practically feasible without restrictions. A solution can be the determination of the influencing factors of the park cooling effect (PCE), and then optimizing existing parks or planning new urban parks considering the importance of the influencing factors to maximize the PCE, thus effectively mitigating the UHI [18,19].

Research on the PCE has primarily employed qualitative and quantitative methods [18,19,20,21,22,23]. When exploring the spatial distribution of land surface temperature (LST), certain urban areas have places with low temperatures, corresponding to ecological infrastructure, such as parks, rivers, green spaces, and lakes [20]. Based on a qualitative analysis, researchers have used cooling indicators to quantify the PCE [20,21,22,23]. The difference between the temperature of the green space and that of the surrounding areas of the park is defined as the park cooling intensity (PCI), and the range of surrounding areas affected by green space is defined as the park cooling distance [23]. However, the PCE is significantly different in different cities [21,22,23,24]. For example, the average cooling intensity of green spaces in Fuzhou was determined to be 1.78 °C [23]; the cooling intensity of 68 green spaces in Shanghai was between 0.78 °C and 5.20 °C, with an average value of 3.02 °C [21]; the average PCI and park cooling area (PCA) of 54 parks in Wuhan were 3.5 °C and 131.6 ha, respectively [22]; in Xi’an, the average value of the PCA was 63.62 ha [24]. These conclusions were derived from previous studies on the PCE in the case of individual cities; however, the national spatial pattern of the PCE remains unclear [25]. In addition, case studies on the PCE have been conducted only considering individual cities, where the number and types of urban parks may be limited [20,26,27]. Therefore, the PCE must be investigated on a larger spatial scale as large sample studies may provide more general conclusions, which may better support urban park planning towards the mitigation of the UHI.

The PCE is affected by many factors, such as park patch characteristics, landscape composition, surrounding environment, and background climate [19,28,29,30,31]. The park area can explain more than 50% of the variation in the PCE in eastern China and is thus usually identified as the dominant factor [19]. The PCI is typically positively correlated with the complexity of geometry of parks, and thus the cooling effect of parks with complex shapes may be more significant [32,33]. Landscape composition within parks is also a crucial factor of the PCE, and waterbodies within parks are the major contributors to the PCE in Wuhan [34]. Further, the surrounding environment of the park has also been confirmed to be related to the PCE. The change in building areas around the park can enhance or weaken the diffusion of cold air in the park [35]. Moreover, background temperature may also be an important influence factor of the PCE [24,36]. Compared to the weather conditions in the case of moderate temperature, parks exhibit a better cooling effect in extreme high temperature weather conditions [24]. Currently, there is a certain understanding of the relationship between the PCE and influencing factors [37,38,39]; these factors have rarely been studied comparatively. Consequently, the factors that exert a stronger influence on the impact of the PCE have not been clearly identified. Thus, it is difficult to determine the priority of planning decisions in urban park planning, particularly when all influencing factors cannot be considered simultaneously.

The existing research has primarily focused on analyzing the PCE in a single city and subsequently discussed the relationship between the PCE and influencing factors, which has a certain guiding role in the case of urban park planning [19,20,40,41,42,43]. However, the number and types of urban parks in a single city may have limitations, and studies on the PCE based on climatic regions are scarce [19]. Meanwhile, the importance of the influencing factors of the PCE is still unclear. The identification of the main factors of the PCE is of great significance for urban park planning. Thus, we selected 502 urban parks in 29 cities in China to perform a comprehensive analysis. The main aims of this study were: (1) quantification of the PCE and analysis of the spatial heterogeneity of PCE; (2) exploration of the relative importance of the influencing factors of the PCE, and subsequent identification of the main factors of the PCE based on climate regions; and (3) to propose suggestions for the optimization or planning of urban parks to alleviate the UHI according to the main influencing factors of the PCE. The research results hope to enrich the understanding of the PCE in different climatic regions and assist urban park planning decisions by maximizing the cooling effect to mitigate the UHI effect.

## 2. Materials and Methods

A flow chart illustrating the methodology is shown in Figure 1. The specific details of the materials and methods are illustrated in the subsequent sections.

### 2.1. Study Area and Data Source

#### 2.1.1. Study Area

With the rapid urbanization process, the UHI is becoming increasingly severe, particularly in provincial capital cities with a high degree of urbanization [44,45]. Thus, urgent measures are required to alleviate the UHI. In this study, 29 cities in China were selected based on the following four principles: (1) a high degree of urbanization with a prominent thermal environment problem; (2) relatively complete park infrastructure in the cities; (3) located in different climatic regions in China and are typical and representative; (4) availability of remote sensing images of Landsat with little or no cloud cover in the urban area during the summer months. Of the selected cities, 9, 9, and 11 were in arid/semi-arid, semi-humid, and humid climates, respectively, according to the Köppen–Geiger climate classification (Figure 2).

#### 2.1.2. Data Source

The spatial resolution of the Landsat remote sensing image is 30~120 m, and the high spatial resolution can be used to study the cooling effect of urban parks [20,46]. Twenty-eight Landsat images from summer of 2019 or 2020 were obtained from the United States Geological Survey website (USGS, https://earthexplorer.usgs.gov (accessed on 20 March 2022)) (Appendix A). During screening, little or no cloud coverage above the city in the image was considered a vital requirement. Precipitation data in China were provided by the National Earth System Science Data Center (http://www.geodata.cn (accessed on 20 June 2022)) [47]. Furthermore, land cover data were maintained by the Global Fine Land Cover Data Product (GLC_FCS30-2020), which can be obtained from the platform of the Earth Big Data Science Engineering Data Sharing Service System (https://data.casearth.cn (accessed on 1 May 2022)) [48]. In this study, land cover data of each city were obtained through mosaic, cutting, projection, and reclassification pretreatment based on this data. Further data information is provided in Appendix A.

### 2.2. Urban Parks and Influencing Factors

In this study, urban parks were extracted via artificial visual interpretation using ArcGIS 10.7 based on high-spatial-resolution Google Earth images. Certain principles were followed for the urban park selection process: (1) the main types of land cover in park should be vegetation or waterbody; (2) as the spatial resolution of LST is 30 m, the area of the selected parks should be greater than 0.09 ha; (3) parks with significant differences in area and shape should be selected; (4) the selected urban park should share a boundary with a gray landscape. Based on the above principles, 502 urban parks were selected as research samples in this study, of which 102, 185, and 215 were in arid/semi-arid, semi-humid, and humid climates, respectively. Further, details regarding the selected parks are provided in Appendix A.

Previous studies have revealed the relationship between the PCE and various influencing factors [18,19,36,41,49]. In this study, 10 influencing factors that have been widely used in past studies were selected, and no strong correlation existed between any two influencing factors. These influencing factors can be divided into four categories: park patch characteristics, waterbody characteristics within parks, surrounding environment of parks, and surrounding climate of parks. The main factors of the PCE were identified via the quantification of the contribution rate of the influencing factors. Table 1 lists the definitions and calculations of the influencing factors.

### 2.3. Methods

#### 2.3.1. Land Surface Temperature (LST) Retrieval

Previous studies have proven that among the many algorithms for land surface temperature (LST) retrieval, the radiative transfer equation (RTE) exhibits the highest accuracy and is the most widely used [23,50]. The principle of the RTE involves the estimation of the effects of atmosphere on surface heat radiation owing to the total heat radiation observed by satellite sensors, and then obtaining the surface thermal radiation intensity. Subsequently, the intensity of thermal radiation can be converted into the land surface [51]. In this study, RTE was used to retrieve the LST. The most important formulae involved are:(1)$$L_{\lambda }=\mathrm{gain}\times \mathrm{DN}+\mathrm{offse}$$
where L_λ_ is the surface thermal radiance intensity received by the satellite sensors, DN is the digital number for a given pixel, and gain and offset can be obtained by the header file.
(2)$$B_{\left(T_{S}\right)}=\left[L_{\lambda }-L↑-\tau \left(1-\varepsilon \right)L↓\right]/\varepsilon \tau$$
where B_(Ts)_ is the ground radiance, T_s_ is the LST, L↓ is the atmospheric downward radiance, L↑ is the atmospheric upward radiance, τ is the atmospheric transmissivity, and ε is the given land surface emissivity. L↓, L↑, and τ were acquired after entering the image imaging date and latitude and longitude from the National Aeronautics and Space Administration (http://atmcorr.gsfc.nasa.gov (accessed on 20 March 2022)). Further, τ can be estimated based on the land-use classification [40].

Consequently, the LST can be calculated using Equation (3)
(3)$$T_{s}=K_{2}/\ln\left(K_{1}/B_{\left(T_{S}\right)}+1\right)$$
where for Landsat-7, K_1_ = 666.09 W/(m^2^·sr·μm), K_2_ = 1282.71 W/(m^2^·sr·μm).

For Landsat-8 TIRS band 10, K_1_ = 774.89 W/(m^2^·sr·μm), K_2_ = 1321.08 W/(m^2^·sr·μm).

To assess the LST retrieval accuracy, we compared the air temperature from weather stations in 29 cities with the retrieved land surface temperature [52], which found that the change trend of the air temperature data basically aligned with the LST. In addition, the land surface temperature in cities was usually higher than the air temperature, which is consistent with the actual situation (Appendix A). Thus, the retrieved land surface temperature can be used in this study.

#### 2.3.2. Calculation of Park Cooling Effect (PCE)

A MultipleRing buffer analysis has been widely used to quantify the PCE in previous studies [19,20,22]. According to the 30 m resolution of Landsat images, 10 buffer rings with a width of 30 m were established from each park boundary [20]. Further, the distance from the park boundary to the buffer ring was set as the independent variable r, and the average LST in each buffer ring was set as the dependent variable T. Previous studies showed that the cubic polynomial was the most suitable for describing the relationship. In this study, the relationship of T_(r)_ was established as follows:(4)$$T_{\left(r\right)}={\mathrm{ar}}^{3}+{\mathrm{Br}}^{2}+\mathrm{cr}+d$$

According to the Law of Diminishing Marginal Utility [53], with an increase in the distance from the park boundary, the LST in the buffer ring increases; however, the rate of increase continually decreases until it becomes 0. When the increase is 0, the first derivative of the T_(r)_ function is 0, which is referred to as the first turning point. The distance from the park boundary to the first turning point is defined as the park cooling distance (Figure 3). Within this distance, the park is considered to have the cooling effect, beyond which the park no longer has cooling effect. In this study, first, the location of the first turning point of the PCE was identified using the T_(r)_ function. Subsequently, the park cooling intensity (PCI) and park cooling area (PCA) were selected to quantify the PCE [19,20]. The PCI was defined as the temperature difference between the first turning point and the average temperature inside the park. Further, the PCA was defined as the area of the surrounding area affected by the PCE. After determining the first turning point position of the T_(r)_ function curve to obtain the park cooling distance, the buffer with the park cooling distance was established as a buffer distance for each park by using the buffer analysis tool in ArcGIS 10.7. Thereafter, the buffer area was calculated to obtain the PCA.

#### 2.3.3. Dominance Analysis

The dominance analysis (DA) method can better estimate the relative importance of related independent variables and is one of the successful methods currently used [54]. This is because it yields more accurate and reliable estimation results, while being better suited to the actual situation. In this study, the DA was used to quantify the contribution of influencing factors to the PCE, explore the importance of influencing factors, and identify the main factors of the PCE. When using k for the number of explanatory variables in the model, the original model corresponded to 2^k^ − 1 subset models. First, using a dominance analysis, all subset models were regressed, and the total variance of the linear regression model was decomposed and assigned to each explanatory variable to calculate the relative contribution of each variable. Consequently, the mean value was used as the final contribution rate of the explanatory variables. Subsequently, the explanatory variables were ranked according to the final contribution to determine the importance of the explanatory variables and to identify the main factors [55,56]. Furthermore, the dominance analysis can only determine the importance of each explanatory variable; thus, correlation and regression analyses were also used to explore the relationship between the PCE and the main factors.

## 3. Results

### 3.1. Spatial Heterogeneity of Park Cooling Effect (PCE)

The PCE showed that the LST in the park was lower than the LST in the surrounding area of the park, thereby forming an obvious “cold spot” in the local urban thermal environment. To quantify the PCE using the PCI and PCA, the results showed that the PCE exhibited obvious spatial heterogeneity (Appendix A). The mean cooling intensity of 502 urban parks was 3.44 °C, with values ranging within 0.22–8.54 °C. Donghu Park in Jiayuguan had the greatest cooling intensity of 8.54 °C; the PCE also affected the surrounding areas of the park. Further, the PCA averaged at 39.8 ha, ranging within 3.11–203.13 ha. Forest Plant Park in Harbin and Vanke Community Park in Shenyang (cooling effect on the 3.11 ha area around the park) yielded the greatest and smallest PCA, respectively.

The cooling effect of urban parks in different climatic regions was different. Specifically, in the humid region, the PCE was relatively high, with the average PCI and PCA of 3.58 °C and 44.38 ha, respectively, whereas the PCE (3.22 °C and 32.65 ha) in arid/semi-arid regions was significantly lower (Figure 4). Owing to the difference in the number and area of parks in each city, the cities could be ranked directly in terms of the PCE values. However, the PCE in Shenzhen (4.64 °C, 52.35 ha), Nanchang (3.47 °C, 44.49 ha), Shanghai (3.05 °C, 45.82 ha), etc., was significantly higher than that in Hohhot (2.83 °C, 33.97 ha), Ordos (2.29 °C, 25.5 ha), Jiuquan (2.88 °C, 28.32 ha), etc. Overall, the parks in cities in the southeast exhibited a more significant cooling effect than those in cities in the northwest on a national scale (Figure 5).

### 3.2. Dominance of Influencing Factors on Park Cooling Intensity (PCI)

Figure 6 shows the contribution rate of the influencing factors to the PCI with three climatic regions. In arid/semi-arid regions, the waterbody characteristics within parks and the surrounding environment of parks were found to be the main influencing factors of the PCI, with a total contribution rate of 78.02%. The contribution rate of the waterbody characteristics within parks was 41.82% (dominant factor of the PCI), of which the contribution rate of waterbody area ratio (WR) and waterbody aggregation index (WAI) were 16.88 and 15.26%; the importance ranked first and fourth among all factors, respectively. The contribution rate of the surrounding environment of parks was 36.2%, of which the contribution rates of the buffer_green_rate (BGR) and buffer_imperious_rate (BIR) were 15.66% and 15.27%, respectively. These were inferior only to the WR and background temperature (BGT), and they were also key factors of the PCI. The importance of park area (PA) and landscape shape index (LSI) to the PCI was far less than the other influencing factors, owing to a contribution rate of only 3.09% and 2.62%, and thereby ranking eighth and ninth among all factors.

In semi-humid regions, waterbody characteristics within parks and the surrounding climate of parks were the main influencing factors of the PCI, with a total contribution rate of 81.31%. In contrast to arid/semi-arid regions, the importance of the surrounding environment of parks to the PCI decreased significantly. The park surrounding climate became the dominant factor of the PCI, with the contribution rate of 42.94%, of which the contribution rate of the BGT was 37.66%, and the importance ranked first among all factors. The contribution rate of the waterbody within parks was 38.37%, of which the contribution rates of the WR and waterbody edge density (WED) were 18.63% and 11.71%. They ranked second and third among all factors, and they were only inferior to the BGT. The remaining factors were less important to the PCI, with the contribution rate not exceeding 10% (Figure 6).

In humid regions, the contribution rate of the waterbody characteristics within parks, surrounding climate of parks, and park patch characteristics all exceeded 20%, which were the main influencing factors of the PCI, with a total contribution rate of 91.99%. Among them, the contribution rate of the surrounding climate of parks was 37.28%, which was the dominant factor of the PCI, of which the contribution rate of the BGT was 34.02%, thus ranking first among all the factors. Further, the contribution rate of waterbody characteristics within parks was 33.13%, of which the contribution rate of the WR was 22.16%, which is second only to the BGT. In contrast to the other climatic regions, in the humid region, the contribution rate of park patches was 21.58%, of which the contribution rates of the PA and LSI were 11.18% and 10.4%, respectively, thereby ranking as third and fourth among all factors (also important factors of the PCI) (Figure 6).

### 3.3. Dominance of Influencing Factors on Park Cooling Area (PCA)

As shown in Figure 7, park patch characteristics were the dominant factor of the PCA in the three climatic regions, and the contribution rate was approximately 80%. However, the importance of the PA and LSI in park patch characteristics to the PCA differed under different climate regions. In arid/semi-arid and semi-humid regions, the PA was the dominant factor of the PCA, with a contribution rate of 60.28% and 54.63%, respectively, ranking the first among all factors. However, the contribution rates of the LSI were 17.42% and 24.13%, second only to the PA. However, in the humid region, the PA and LSI were almost equally important to the PCA, with the contribution rates of 38.64% and 40.13%, respectively, (contribution rate of the LSI was slightly higher than that of the PA, which was the dominant factor of the PCA). The remaining factors were not important to the PCA, and the contribution rate did not exceed 10% in the three climate regions (Figure 7). The PA and LSI were significantly positively correlated with the PCA (Appendix A), implying that increasing the park area and the complexity of the park boundary shape were effective measures to increasing the PCA in three different climatic regions.

## 4. Discussion

### 4.1. Impact of Buffer Range on Park Cooling Effect

The buffer zones analysis method was usually used to quantify the PCE in previous studies. However, the size of the buffer zone may affect the quantification of the PCE. Previous studies have used different sizes of buffer zone, such as 300, 500, 900, 990, and 1200 m, which were usually divided into approximately a 10~40 buffer ring with a width of 30 m to form the temperature curve [19,20,22,26,57]. Further, certain studies have used buffer widths based on the length of the park radius [58]. However, two questions entail further discussion. Does the size of the buffer zone affect the quantization of the PCE? If yes, then what size buffer should be used to quantify the PCE?

In this study, we used a buffer zone of 300 m width to quantify the PCE. In the early stage of data processing, certain parks were randomly selected from a sample of 502 urban parks and analyzed for the PCE using the 300, 600, and 900 m buffer width. As shown in Figure 8, the size of the buffer zone directly affects the quantification of the PCE. On the one hand, from the perspective of mathematical statistics, the larger the buffer width, the poorer the statistical description (R^2^) of the distance–temperature curve fitting relationship. When the buffer zone width was 300 m, the correlation coefficient (R^2^) of the fitted curve was the best [20]. On the other hand, from the perspective of the description of the PCE, considering outskirts from the boundary of the park, the LST gradually increased with the increase in the distance from the park; however, the increase rate decreased until it tended to 0 along the further distance. Thus, the increase of 0 was referred to as the first turning point in previous studies. Consequently, the accurate identification of the location of the first turning point is crucial to quantify the PCE. According to mathematical knowledge, the point must be close to the temperature turning interval in the fitting curve. The position of the first turning point identified by the 300 m buffer zone was within or closer to the temperature turning interval (between the two red points in Figure 8), while that of the first turning point identified by the 600 and 900 m buffer was further away from this interval. Therefore, we deem that the optimal buffer zone width of quantifying the PCE is 300 m and using 600 and 900 m buffers will expand the park cooling distance, thus overestimating the PCA and increasing the uncertainty of the PCI.

### 4.2. Park Cooling Effect and Main Influencing Factors

The cooling effect of urban parks reduces the temperature inside the park and in a certain area around the park, which is considered to be an important measure to alleviate the UHI. At present, the PCE been extensively studied from an urban perspective. However, studies on the cooling effect of urban parks in different climatic regions are scarce. Based on the PCE across different climatic regions, those in arid/semi-arid regions were found to be significantly lower from those of other climates, which is consistent with previous research results [19,25]. From a nationwide perspective, the urban parks in the southeast China generally exhibited a more significant cooling effect than those in the northwest China. This may be attributed to the relatively imperfect nature of the urban park facilities, small park area, and limited vegetation coverage and waterbody area within parks in northwest China, which results in an overall low PCE [25].

By studying the importance of influencing factors, the factors exerting a greater influence on the PCE can be identified. Consequently, optimizing the existing parks or planning new parks according to the importance of influencing factors can aid in obtaining more significant cooling effects [18]. The study found that, in the arid/semi-arid regions, waterbodies within parks were the dominant factor of the PCI. Whereas, in the semi-humid and humid regions, the surrounding climate of parks was the dominant factor of the PCI, with waterbodies within parks ranking second among all factors. These are similar to the previous research conclusion in Wuhan, where they considered that the waterbody is the most important factor of the PCI [34]. However, Geng et al. (2022) found that the park area was the dominant influencing factor of the PCE, accounting for more than 50% of the PCI variation [19]. We concluded that there may be two reasons for the difference in results. (1) The study area of Geng et al. (2022) is in the east of China, which is part of the humid region of this study. Thus, differences in the selection of study areas and park samples may lead to differences in study results. (2) Geng et al. (2022) selected urban parks whose areas were mainly less than 5 ha, and parks with small areas were less likely to have waterbodies, which may result in the park area being the dominant factor of the PCI (Appendix A). In this study, the importance of the park area to the PCI ranked third among all factors in the humid region, being inferior to only the WR and BGT. In addition, after selecting the urban parks without waterbodies, the dominance analysis of the influencing factors of the PCI was conducted again. The results showed that the park area was the dominant factor of the PCI, with a contribution rate of 59.25% (Appendix A). Meanwhile, the results of the PCE study based on climate region were different from those of previous single city research, which is very necessary for enriching the understanding of the PCE and guiding urban park planning.

The importance of influencing factors of the PCE differed in the different climate regions [19,25]. Waterbodies within urban parks played a more significant role in the PCE in arid/semi-arid regions. However, from arid/semi-arid, semi-humid to humid regions, the importance of the WR on the PCI gradually increased, contrary to the trend of the total contribution rate of waterbody characteristics within parks. This indicates that the WAI and WED are also important factors for the PCI in arid/semi-arid regions. In addition, in arid/semi-arid regions, the importance of the surrounding environment of parks to the PCI was second only to the waterbody within parks, and far exceeded that of the semi-humid and humid regions. Therefore, adjusting the surrounding environment of parks is also an effective measure to enhance the PCI in arid/semi-arid regions. Park patch characteristics were more important for the PCI in the humid region, by optimizing the existing parks or planning new parks according to the importance of influencing factors to the PCI in different climatic regions, which could effectively increase the PCI. However, for the PCA, the park patch characteristics were the dominant factor, and the importance was the same in different climatic regions. However, from arid/semi-arid, semi-humid to humid regions, the importance of the PA to the PCA decreased gradually, while that of the LSI to the PCA increased significantly. Even in the humid regions, the contribution rate of the LSI was 40.13%, which exceeded the PA, and was thus the dominant factor of the PCA. To clarify the importance of factors influencing the PCE in different climate regions, which is of great significance to the planning of urban parks, urban planners should assign higher priority to factors that have a more important impact on the PCE, and thus maximize the PCE to fully mitigate the UHI.

### 4.3. Implications for Urban Park Planning

#### 4.3.1. The Optimal Proportion of Waterbody and Vegetation Area within Parks

In the arid/semi-arid regions, the WR is the most important factor for the PCI. Whereas, in the semi-humid and humid regions, although the BGT is the dominant factor for the PCI, the enhancement of the PCE to alleviate the UHI by improving the BGT is not feasible. Following the BGT, the WR is the most important factor for the PCI. Thus, increasing waterbodies within parks is an effective measure to enhance the PCI; however, a larger waterbody area in the park does not imply that it is better [59]. As shown in Figure 9, in arid/semi-arid and semi-humid regions, when the proportion of waterbody and vegetation area within the park is approximately 1:2 and 1:1, respectively, the PCI is the largest. If the area of the waterbody continues to increase at this time, the PCI will remain stable or even decline. In the humid region, the proportion of waterbody and vegetation area within a park is approximately linear to the PCI (Figure 9). Theoretically, when planning urban parks in humid areas, the waterbody area within a park can be increased to the best extent possible to enhance the PCI. In addition, waterbodies within a park should avoid dispersion, and the shape of the water boundary should be as complex as possible (Appendix A).

#### 4.3.2. Threshold Value of Park Area

Park patch characteristics are the dominant factors of the PCA. Increasing the park area and the complexity of the park boundary shape are effective measures to increase the PCA (Appendix A). However, the PCA exhibits a nonlinear increase with the increase in the park area. This nonlinear relationship makes the park area possibly have a critical value, which was defined as the threshold value of the park area in previous studies [19,49]. As shown in Figure 10, in arid/semi-arid and semi-humid regions, the threshold value of the urban park area is approximately 16 ha, whereas in humid regions, it is approximately 19 ha. When the park area changes within the threshold value, the impact of the park area on the PCA is significant; once the park area increases above the threshold value, the impact of the park area change on the PCA gradually weakens [27,60]. Thus, building urban parks with a corresponding area size and complex boundary shape in different climate regions can help increase the PCA, serve more urban residents, and achieve the optimal value from the perspective of cost effectiveness.

### 4.4. Limitations and Future Research Directions

In this study, we quantified the cooling effect of urban parks, and used the dominance analysis to study the relative importance of influencing factors of the PCE based on climate region. The study results can provide guidance for the planning and designing of urban parks to enhance the cooling effect to alleviate the UHI. However, this study also has certain limitations. First, a total of 28 remote sensing images of Landsat were used for LST retrieval in this study. The application of multi-temporal remote sensing images may affect the consistency of the LST for different urban parks, and thus may impact the results to a certain extent [19]. Second, the influencing factors of the PCE were not fully considered. For example, there are a few impervious surfaces within parks, such as roads and rest places; however, it was difficult to identify them in this study [21,26]. Further, humidity, wind speed, etc., may also affect the PCE [26,61]; however, these factors were not considered in this study. Third, this study only focused on the cooling effect of urban parks during the daytime in summer. The PCE should not be studied only in one season, and the diurnal variation and seasonal comparison will be the next research direction [62,63].

## 5. Conclusions

Exploring the importance of influencing factors on the PCE in different climate regions can provide guidance for urban planning. Optimizing the existing parks or planning new parks according to the importance of influencing factors to the PCE will maximize the cooling effect to mitigate the UHI. Our findings suggest that the PCE increased from arid/semi-arid, semi-humid to humid regions. For the PCI, in arid/semi-arid regions, waterbodies within parks and the surrounding environment of parks were the main factors; in the semi-humid region, they were waterbodies within parks and the surrounding climate; and finally in the humid region, the waterbody within parks, surrounding environment, and surrounding climate of parks were the main factors. For the PCA, park patch characteristics were the dominant factors, with a contribution rate of approximately 80% in each climatic region. Waterbodies within parks and the surrounding environment of parks are crucial to the PCE in arid/semi-arid regions; however, the park patch characteristics are more important in the humid regions. The addition of waterbodies within parks is an effective measure to enhance the PCI. However, the optimal area proportions of waterbody and vegetation within parks are approximately 1:2 and 1:1, respectively, in arid/semi-arid and semi-humid regions. Thus, the maximum possible waterbody area within parks can be added to enhance the PCI in the humid region. Increasing the park area can increase the PCA. The threshold park area value was 16 ha in arid/semi-arid and semi-humid regions, and 19 ha in humid regions. Furthermore, increasing the complexity of the park boundary shape can also increase the PCA, particularly in the humid region. Thus, these findings are expected to provide guidance for urban park planning and management for different climate regions to a certain extent to mitigate the UHI.

## Figures and Tables

**Figure 1 ijerph-19-15496-f001:**
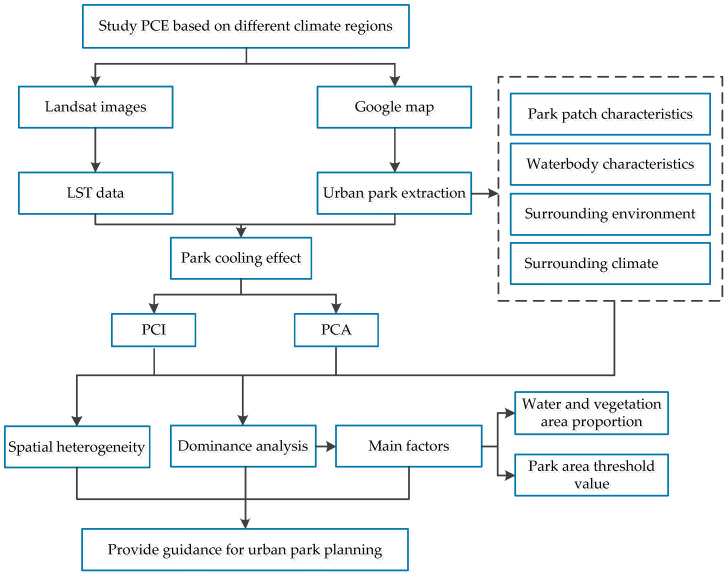
Technical flow chart.

**Figure 2 ijerph-19-15496-f002:**
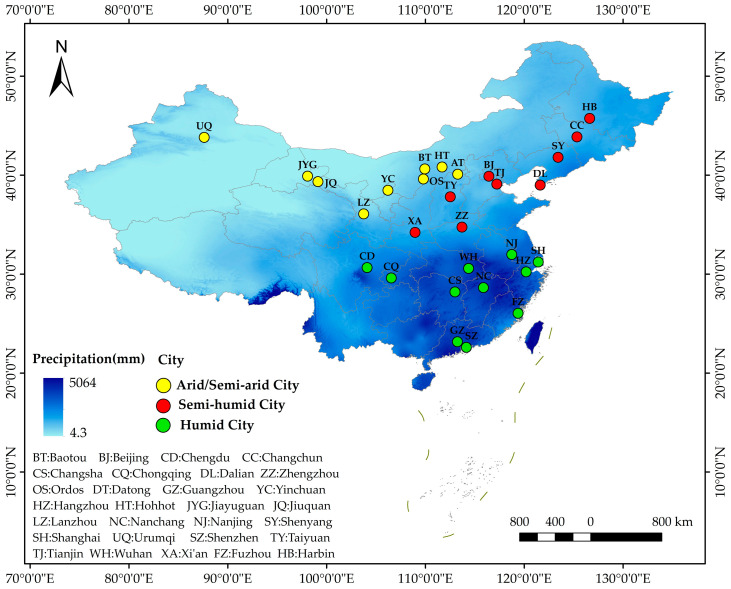
Locations of the 29 selected cities with three different climatic regions in China.

**Figure 3 ijerph-19-15496-f003:**
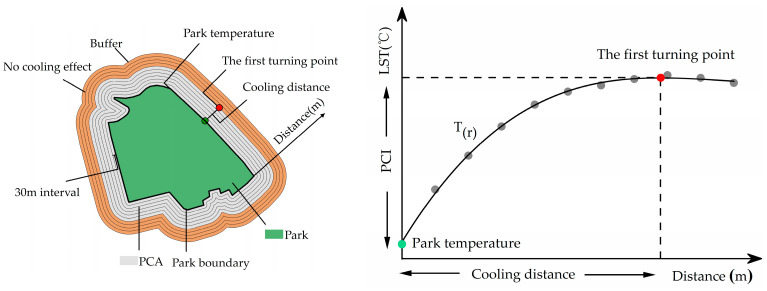
Sketch map of quantification method of park cooling effect.

**Figure 4 ijerph-19-15496-f004:**
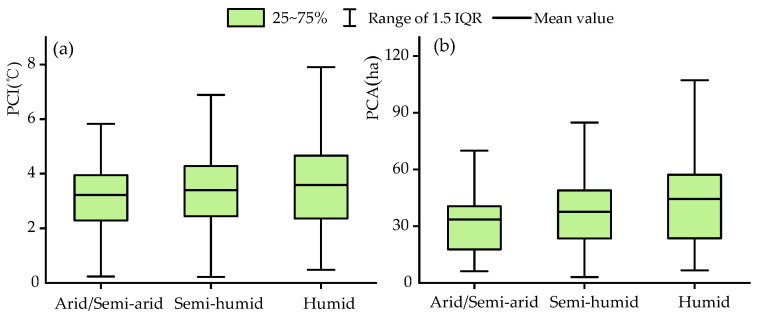
Differences of urban park cooling effects in different climatic regions: (**a**) is PCI, (**b**) is PCA.

**Figure 5 ijerph-19-15496-f005:**
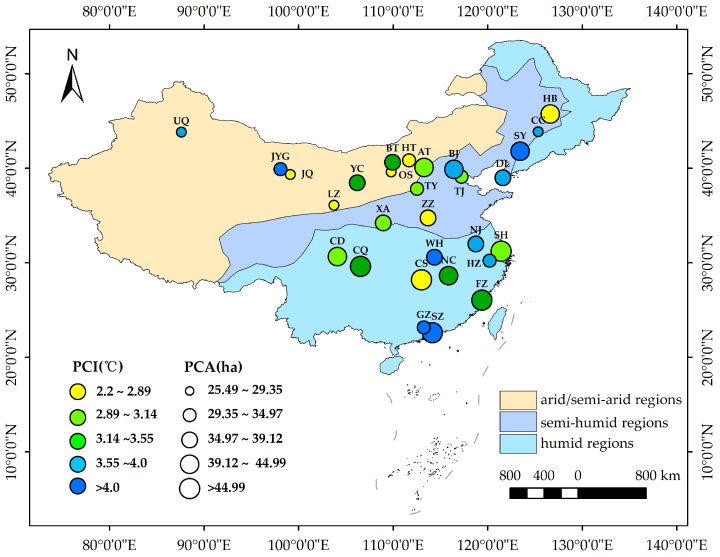
The national spatial pattern differences of the cooling effect.

**Figure 6 ijerph-19-15496-f006:**
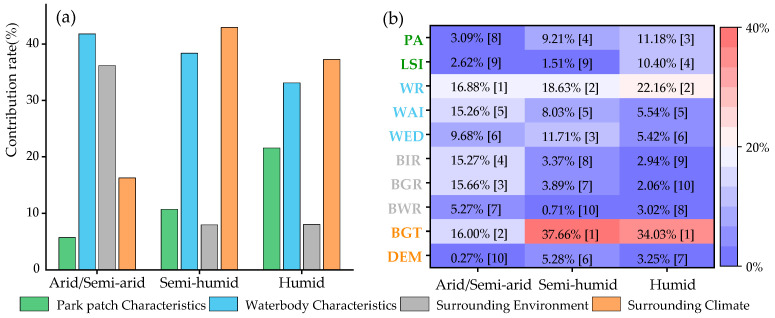
Contribution rate of influencing factors of park cooling intensity: (**a**) is the total contribution rate of the four categories, (**b**) is the individual contribution rate of 10 impact factors, [] is the importance ranking of impact factors.

**Figure 7 ijerph-19-15496-f007:**
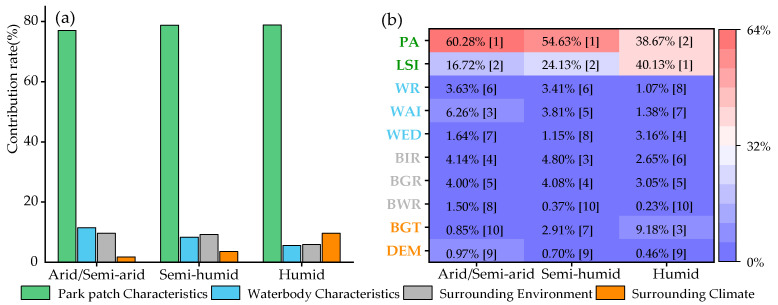
Contribution rate of influencing factors of park cooling area: (**a**) is the total contribution rate of the four categories, (**b**) is the individual contribution rate of 10 impact factors, [] is the importance ranking of impact factors.

**Figure 8 ijerph-19-15496-f008:**
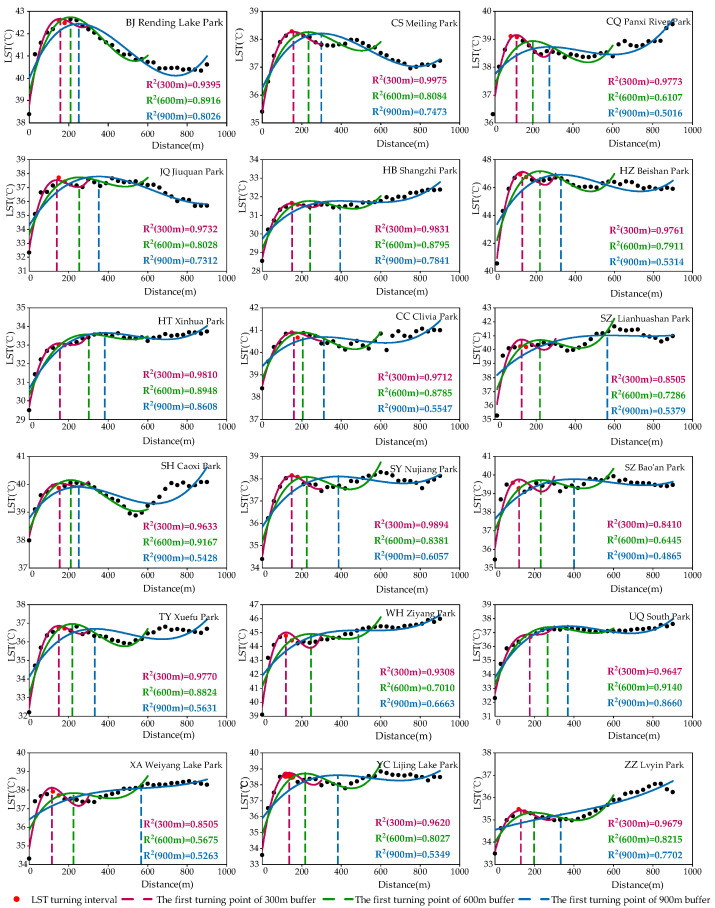
Impact of buffer zone width on cooling effect in randomly selected urban parks.

**Figure 9 ijerph-19-15496-f009:**
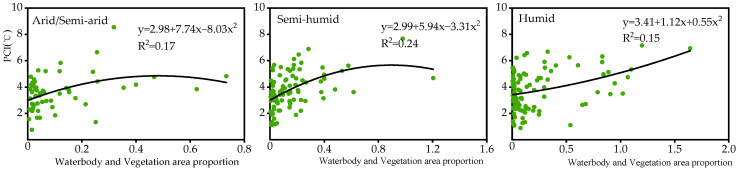
Relationship between PCI and the proportion of waterbody and vegetation area within parks in three climatic regions.

**Figure 10 ijerph-19-15496-f010:**
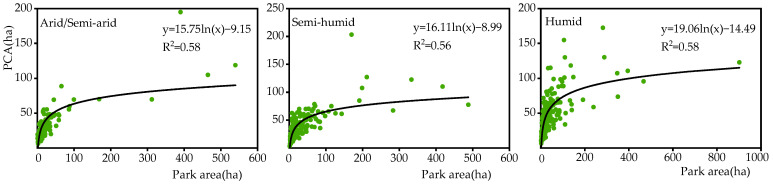
Relationship between PCA and park area in three climatic regions.

**Table 1 ijerph-19-15496-t001:** Information on influencing factors on cooling effect of urban parks.

Categories	Impact Factors	Formula and Range	Definition
Park patch characteristics	Park area (PA)	≥0.09 ha	The area of an urban park
	Landscape shape index (LSI)	$\mathrm{LSI}=\frac{P}{2\sqrt{A\times \pi }}$ LSI > 1	The landscape shape index of an urban park
Waterbody characteristics within parks	Water area ratio (WR)	$\mathrm{PWR}=\frac{PWA}{PA}$ 1 ≥ PWR ≥ 0	The proportion of the waterbody in an urban park
	Waterbody aggregation index (WAI)	$\mathrm{WAI}=\left[\frac{g_{ii}}{max\to g_{ii}}\right]$(100) 100 ≥ WAI ≥ 0	Proximity of waterbody patches in an urban park
	Waterbody edge density (WED)	$\mathrm{WED}=\frac{\mathrm{WP}}{\mathrm{WA}}\times 10^{6}$ ED ≥ 0	Edge length between heterogeneous landscape patches on waterbody per unit area
Surrounding environment of parks	Buffer_imperious_rate (BIR)	$\mathrm{BIR}=\frac{BIA}{BA}$ 1 ≥ BIR ≥ 0	The proportion of the impermeable surface in the 300 m buffer of an urban park
	Buffer_green_rate (BGR)	$\mathrm{BGR}=\frac{BGA}{BA}$ 1 ≥ BGR ≥ 0	The proportion of the green area in the 300 m buffer of an urban park
	Buffer_water_rate (BWR)	$\mathrm{BWR}=\frac{BWA}{BA}$ 1 ≥ BWR ≥ 0	The proportion of the waterbody area in the 300 m buffer of an urban park
Surrounding climate of parks	Background temperature (BGT)	>0	Average land surface temperature in the 300 m buffer of an urban park
	DEM	≥0	Average elevation of an urban park

## Data Availability

The Landsat data we used are available at the U.S. Geological Survey (https://earthexplorer.usgs.gov/ (accessed on 20 March 2022)). The Google Earth images we used are available at the Bigemap GIS Office. The precipitation data we used are available at the National Earth System Science Data Center (http://www.geodata.cn/ (accessed on 20 June 2022)). The land cover data we used are available at the Earth Big Data Science Engineering Data Sharing Service System (https://data.casearth.cn/ (accessed on 1 May 2022)). These data are all publicly available.

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
