# Peer review of "Dominance of Influencing Factors on Cooling Effect of Urban Parks in Different Climatic Regions"

_ijerph, 2022, doi:10.3390/ijerph192315496_

Round 1
Reviewer 1 Report
Comments:
The manuscript studied the factors influencing the park cooling effect of parks(PCE) and quantified the PCE based on the park cooling intensity (PCI) and park cooling area (PCA). Moreover, this study compared the main factors for different climate regions. These results may provide some effective suggestions for urban planning in the future.
The overall logic of the manuscript is clear and smooth. Moreover, some other details in the article need to be modified. This reviewer will report them below. This reviewer hopes the comments could be constructive and helpful in improving this interesting work can eventually be published after revision.
---Introduction:
1. Lines 57.” Research on PCE has primarily employed qualitative and quantitative methods[18-23]…Based on a qualitative analysis, researchers have used cooling indicators to quantify PCE [20-23]…”
Could the author elaborate on the qualitative and quantitative methods for quantifying PCE?
2. There are many abbreviations in this manuscript. It could be more readable if the authors provided an abbreviation list.
3. “The urban population in China is expected to increase by 255 million [12], which will further aggra- vate the UHI effect” How do you get this data of “255 million urban population in China”?? How large population at present and when it will increase by 255 million? Please clarify it clearly otherwise it will be misleading.
4. Actually urban parks have several positive impacts, for instance, not only benefit reducing urban heat island intensities, but also generally improve urban ventilation and reduce urban pollutant dispersion (e.g. Sha et al. The impact of urban open space and ‘lift-up’ building design on building intake fraction and daily pollutant exposure in idealized urban models. Science of The Total Environment 2018, 633:1314-28.) and provide space for public activities and animal livings etc. I suggested to discuss a little about these positive impacts with 1-2 reference papers before the discussion on its impacts on urban heat island.
---2. Materials and Methods:
1. Page 3, Figure 1: Could the author give more details about the “Surrounding environment”? The environment is a very complex concept. How do the author distinguish the difference between the surroundings?
--- 3. Results:
1. Page 7-8, Lines 246-254,” The cooling effect of urban parks in different climatic regions was different. Specifically, in humid region, the PCE was relatively high, with the average PCI and PCA of 3.58 °C nd 44.38 ha respectively, whereas the PCE (3.22°C and 32.65 ha) in arid/semi-arid regions was significantly lower (Figure 4). Owing to the difference in the number and area of parks in each city, the cities could be ranked directly in terms of the PCE values. However, the PCE in SZ (4.64°C, 52.35 ha), NC (3.47°C, 44.49 ha), HS (3.05°C, 45.82 ha), etc, was significantly higher than that in HT (2.83°C, 33.97 ha), OS (2.29°C, 25.5 ha), JQ (2.88°C, 28.32 ha), etc. Overall, the parks in cities in southeast exhibited a more significant cooling effect than those in cities in northwest on a national scale (Figure 5).”
There are too many abbreviations. I suggest to use the full name of NC, HS, HT, OS, JQ etc. Otherwise it is difficult to read these analysis.
1) First,” in southeast” should be “in the southeast,” and ” in northwest” should be “in the northwest.”There are many other similar grammatical problems in this paper. Please double-check and revise them.
2) In this section, the authors analyze the differences in the PCE in different climate regions, but when analyzing the number of parks on the cooling effect of research, would it be better to mark the climate type behind the city? Alternatively, identify the climate type to which different cities belong on the map.
Reviewer 2 Report
Thank you for possibility to revise the manuscript. The manuscript deals with very important issue of enhancing the cooling effect of urban parks. The text flows very well and it has all important features of scientific paper. I suggest it to be published after minor revisions, mainly after the small text corrections.
L22 (+ L271) – you use here the concept “sub-humid”, but in the text further, you use “semi-humid”, be consistent in using the terms
L123 – Figure 2 – you present here by different colors cities in different climate regions. It would be good to show this categorization somehow also in Figure 5
L157 – Table 1 – please differentiate the individual categories of impact factors, for example by lines or by colors, as you have it in further figures (6 or 7).
L213 – Please explain what does the orange area mean.
L228 – put space between “factors[55,56]”
L250 - 251 – please write somewhere the citation of Figure 2, where the abbreviations of cities are present
L257 - put space between “…intensity(PCI)”
L292 – Figure 6 – make green the first two indexes of impact factors PA and LSI in figure 6b
L309 – I think you meant PCA in the sentence “…measures to increasing PA in three …”.
L336 - put full stop at the end of a sentence “…studies. Consequently…”and correct the capital letter
Reviewer 3 Report
Please find the attachment.
